# Complementary evolution of coding and noncoding sequence underlies mammalian hairlessness

**Amanda Kowalczyk[1,2], Maria Chikina[2], Nathan Clark[3]***

[1]Carnegie Mellon-University of Pittsburgh PhD Program in Computational Biology, Pittsburgh, United States; [2]Department of Computational Biology, University of Pittsburgh, Pittsburgh, United States; [3]Department of Human Genetics, University of Utah, Salt Lake City, United States

**Abstract** Body hair is a defining mammalian characteristic, but several mammals, such as whales, naked mole-rats, and humans, have notably less hair. To find the genetic basis of reduced hair quantity, we used our evolutionary-rates-based method, RERconverge, to identify coding and noncoding sequences that evolve at significantly different rates in so-called hairless mammals compared to hairy mammals. Using RERconverge, we performed a genome-wide scan over 62 mammal species using 19,149 genes and 343,598 conserved noncoding regions. In addition to detecting known and potential novel hair-related genes, we also discovered hundreds of putative hair-related regulatory elements. Computational investigation revealed that genes and their associated noncoding regions show different evolutionary patterns and influence different aspects of hair growth and development. Many genes under accelerated evolution are associated with the structure of the hair shaft itself, while evolutionary rate shifts in noncoding regions also included the dermal papilla and matrix regions of the hair follicle that contribute to hair growth and cycling. Genes that were top ranked for coding sequence acceleration included known hair and skin genes *KRT2*, *KRT35*, *PKP1*, and *PTPRM* that surprisingly showed no signals of evolutionary rate shifts in nearby noncoding regions. Conversely, accelerated noncoding regions are most strongly enriched near regulatory hair-related genes and microRNAs, such as *mir205*, *ELF3*, and *FOXC1*, that themselves do not show rate shifts in their protein-coding sequences. Such dichotomy highlights the interplay between the evolution of protein sequence and regulatory sequence to contribute to the emergence of a convergent phenotype.

*For correspondence:
nclark@utah.edu

**Competing interest:** The authors declare that no competing interests exist.

## Editor's evaluation

Several mammal species, including dolphins, have evolved to be relatively "hairless". In this important work, Kowalczyk and colleagues scan the genomes of multiple species to identify genomic regions that appear to have evolved at a faster or slower evolutionary rate along hairless lineages. Using convincing analyses, they identify a number of protein-coding genes as well as noncoding regions that might explain how hairlessness evolved in mammals. This study is of interest to those investigating the development of the skin and its appendages as well as evolutionary biologists, especially those investigating instances of convergent evolution and those developing phylogenomic methods for genome comparisons.

**eLife digest** Whales, elephants, humans, and naked mole-rats all share a somewhat rare trait for mammals: their bodies are covered with little to no hair. The common ancestors of each of these species are considerably hairier which must mean that hairlessness evolved multiple times independently. When distantly related species evolve similar traits, it can be interpreted as a certain aspect of their evolution repeating itself. This process is called 'convergent evolution' and may provide insights about how different species were able to arrive at the same outcome. One possibility is that they have undergone similar genetic changes such as turning on or off key genes that play a role in the trait's development.

Kowalczyk et al. set out to identify what genetic changes may have contributed to the convergent evolution of hairlessness in unrelated species of mammals. By looking at the genomes of 62 mammalian species, they hoped to link specific genomic elements to the origins of the hairless trait. The genetic sequences under investigation included nearly 20,000 genes that encode information about how to make proteins, as well as 350,000 regulatory sequences composed of non-coding DNA, which specify when and how genes are activated. This marks the first time genetic mechanisms behind various hair traits have been studied in such a diverse group of mammals.

Using a computational approach, Kowalczyk et al. identified parts of the genome that have evolved similarly in mammalian species that have lost their hair. They found that genes and regulatory sequences, that had been previously associated with hair growth, accumulated mutations at significantly different rates in hairless versus hairy mammals. This indicates that these regions associated hair growth are also related to evolution of hairlessness. This includes several genes that encode keratin proteins, the main material that makes up hair. The team also reported an increased rate of evolution in genes and regulatory sequences that were not previously known to be involved in hair growth or hairlessness in mammals. Together these results suggest that a specific set of genetic changes have occurred several times in different mammalian lineages to drive the evolution of hairlessness in unrelated species.

Kowalczyk et al. describe the parts of the genome that may be involved in controlling hair growth. Once their findings are validated, they could be used to develop treatments for hair loss in humans. Additionally, their computational approach could be applied to other examples of convergent evolution where genomic data is available, allowing scientists to better understand how the same traits evolve in different species.

## Introduction

Hair is a defining mammalian characteristic with a variety of functions, from sensory perception to heat retention to skin protection (*Pough et al., 1989*). Although the mammalian ancestor is believed to have had hair, and in fact the development of hair is a key evolutionary innovation along the mammalian lineage (*Eckhart et al., 2008*), numerous mammals subsequently lost much of their hair. Many marine mammals, including whales, dolphins, porpoises, manatees, dugongs, and walruses, have sparse hair coverage likely related to hydrodynamic adaptations to allow those species to thrive in a marine environment (*Chen et al., 2013*; *Nery et al., 2014*). Large terrestrial mammals such as elephants, rhinoceroses, and hippopotamuses also have little hair, likely to enable heat dissipation diminished by the species' large sizes (*Fuller et al., 2016*). Notably, humans are also relatively hairless, a phenotypic characteristic that, while stark, has long been of mysterious origin (*Kushlan, 1980*). Just as hair coverage varies across mammal species, coverage for an individual organism can change over time in response to environmental factors. For example, Arctic mammals such as foxes and hares famously demonstrate dramatic coat changes in different seasons (*Johnson, 1981*).

Hair follicles are established during embryonic development as a result of interactions between epithelial and mesenchymal cells in the skin, and such interactions also drive follicle movement in adults (*Zhou et al., 2018*). Hair follicles consist of a complex set of structures under the skin that support the hair shaft itself, which protrudes above the skin. The hair shaft contains an outer layer called the cuticle, an inner cortex later, and sometimes a central medulla core (*Plowman et al., 2018*). Structures under the skin support the growth and formation of the hair follicle. Of particular interest are the dermal papilla and matrix region, both located at the base of the hair follicle. The dermal

papilla is a key controller of regulation of hair growth and follicle morphogenesis (*Veraitch et al., 2017*). In fact, transplantation of dermal papilla cells has been repeatedly demonstrated to result in hair growth in previously hairless tissue (*Jahoda et al., 1984*; *Jahoda et al., 1993*; *Reynolds and Jahoda, 1992*). Just above the dermal papilla, the matrix generates stem cells to the growing hair shaft and the root sheath (*Plowman et al., 2018*). The two regions work together to regulate and carry out hair growth – the dermal papilla is the master controller that instructs the hair-growing engine of the matrix region.

During hair growth, a hair follicle goes through three stages of growth called anagen, catagen, and telogen phases. During the anagen phase, the hair shaft is generated and grows out through the skin, while catagen phase ends hair growth and telogen phase causes the follicle to become dormant (*Alonso and Fuchs, 2006*).

Changes to several hair-related genes are known to result in hairlessness in specific species. The *Hr* gene in mice, so named because of its role in the hair phenotype, results in hairless mice when knocked out (*Benavides et al., 2009*). In Mexican dogs, the *FOXI3* gene has been found to be associated not only with hairlessness, but also associated with dental abnormalities (*Drögemüller et al., 2008*). In the American Hairless Terrier, mutation in a different gene, *SGK3*, is responsible for relative hairlessness (*Parker et al., 2017*). Fibroblast growth factor genes such as *FGF5* and *FGF7* are also heavily implicated in hair growth because their absence causes drastic changes to coat length and appearance in mice (*Ahmad et al., 1998*). Such genes are associated with keratinocyte growth in which keratins and keratin-associated proteins play a key role. Unsurprisingly, specific structural proteins that comprise hair shafts and their associated genes, known as *KRTAP* genes or hair-specific keratins, are also heavily implicated in hair-related functions (*Plowman et al., 2018*). They also appear to be unique to mammals, although some *KRTAP*-like genes have been found in reptiles (*Eckhart et al., 2008*).

Although genetic changes associated with induced hairlessness in specific domesticated species are useful, it is unclear whether such changes reflect evolutionary changes that result in spontaneous hairlessness and how much such changes are convergent across all or many naturally hairless species. By taking advantage of natural biological replicates of independent evolution of hairlessness in mammals, we can learn about global genetic mechanisms underlying the hairless phenotype.

Mammalian hairlessness is a convergent trait since it independently evolved multiple times across the mammalian phylogeny. We can therefore characterize the nature of its convergence at the molecular level to provide insights into the mechanisms underlying the trait. For example, if a gene is evolving quickly in hairless species and slowly in non-hairless species, that implies that the gene may be associated with hairlessness. We focus on the relative evolutionary rate of genomic sequence, which is a measure of how fast the sequence is evolving relative to its expected rate. Unlike seeking sequence convergence to a specific amino acid or nucleotide, using an evolutionary-rates-based method detects convergent shifts in evolutionary rates across an entire region of interest (such as a gene or putative regulatory element). Evolutionary rate shifts reflect the amount of evolutionary pressure acting on genomic elements, and multiple studies investigating diverse phenotypes have found that phenotypic convergence is indeed associated with convergent changes in evolutionary rates (*Chikina et al., 2016*; *Hiller et al., 2012*; *Hu et al., 2019*; *Kapheim et al., 2015*; *Kowalczyk et al., 2020*; *Partha et al., 2017*; *Partha et al., 2019*; *Prudent et al., 2016*; *Wertheim et al., 2015*). We used RERconverge, an established computational pipeline, to link convergently evolving genes and noncoding regions to convergent evolution of mammalian hairlessness. Previous work using RERconverge (*Kowalczyk et al., 2019*) to detect convergent evolutionary rate shifts in genes and noncoding elements associated with convergently evolving traits has identified the putative genetic basis of the marine phenotype in mammals (*Chikina et al., 2016*), the fossorial phenotype in subterranean mammals (*Partha et al., 2017*; *Partha et al., 2019*), and extreme longevity in mammals (*Kowalczyk et al., 2020*). Those studies revealed trends that are not species-specific, but instead represent relevant genetic changes that occurred phylogeny-wide.

Here, we further explored the genetic basis of hairlessness across the mammalian phylogeny by finding genes and noncoding regions under relaxation of evolutionary constraint (i.e., evolving faster) in hairless species. Such genetic elements likely have reduced selective constraint in species with less hair and thus accumulate substitutions at a more rapid rate. To find genetic elements under accelerated evolution in hairless species, we performed a genome-wide scan across 62 mammal species

using RERconverge on 19,149 orthologous genes and 343,598 conserved noncoding elements. In addition to recapturing known hair-related elements, we also identified novel putative hair-related genetic elements previously overlooked by targeted studies. Importantly, newly uncovered genes and noncoding regions were not only related to keratins, but they also represented a suite of genetic functionality underlying hair growth. Such findings represent strong candidates for future experimental testing related to the hair phenotype.

## Results

### Phenotype assignment

The hairless phenotype in mammals arose at least nine independent times along the mammalian phylogeny (*Figure 1A*, *Figure 1—source data 1*). Genomic regions that experienced evolutionary rate shifts in tandem with mammalian loss of hair were considered potentially associated with phenotype loss. Ten extant and one ancestral hairless species were identified based on species hair density (*Figure 1A*). Broadly, species with skin visible through hair were classified as hairless, namely, rhinoceros, elephant, naked mole-rat, human, pig, armadillo, walrus, manatee, dolphin, and orca. The cetacean (dolphin-orca) ancestor was also included because it was likely a hairless marine mammal.

An ancestral point of phenotypic ambiguity existed at the ancestor of manatee and elephant. Considerable uncertainty exists as to whether the ancestral species had hair and independent trait losses occurred on the manatee and elephant lineages or, alternatively, whether the ancestral species lost hair prior to manatee–elephant divergence and regained hair along mammoth lineages post-divergence (*Roca et al., 2009*). Since foreground assignment of the manatee–elephant ancestor had little impact on skin-specific signal, we retained the parsimonious assignment of the ancestral species as haired with inferred independent losses in the manatee and post-mammoth elephant lineages (*Figure 1—figure supplement 1B*). Similarly, assigning foreground branches based on the state of being hairless or the transition from haired to hairless – that is, assigning the entire cetacean clade as foreground versus only assigning the cetacean ancestor as foreground – had little impact on skin-specific signal (*Figure 1—figure supplement 1A*). In the case of cetaceans, we retained all three branches (orca, dolphin, and the orca-dolphin ancestor) as foreground to maximize statistical power.

### Phenotypic confounders

Hairless species share other convergent characteristics that could confound associations between the hairless phenotype and evolutionary rate shifts. In particular, several hairless species are large and many are marine mammals. Therefore, any signal related to hairless species could be driven instead by confounders. Problems with these two confounders were handled in two different ways.

To handle large body size as a confounder, body size was regressed from relative evolutionary rates on an element-by-element basis. In other words, the residuals from the linear relationship between body size and relative evolutionary rates were retained to eliminate the effect of body size on relative evolutionary rate trend. In doing so, any effects related to the relationship between body size and hairlessness were mitigated.

Marine status, on the other hand, is a trait of potential interest because marine mammals experienced unique hair and skin changes during the transition from a terrestrial to a marine environment. However, it is also of interest how much signal is driven by the marine phenotype versus the hairless phenotype. Therefore, Bayes factors were used to quantify the amount of support for the marine phenotype versus the hairless phenotype. A larger Bayes factor indicated more contribution from one model versus another. A ratio of 5 or greater for the hairless phenotype versus the marine phenotype indicated strongly more support for signal driven by hairlessness. Many hair-related pathways evolving faster in hairless species according to RERconverge also indicated that signal was indeed driven by the hairless phenotype as opposed to its heavy confounder, the marine phenotype, according to Bayes factor analyses (*Figure 2*).

### Species-specific analyses

In addition to conducting convergent evolution analyses to identify genetic elements evolving at different rates across all hairless species, we also conducted complementary analyses to detect elements evolving at different rates in individual hairless species to demonstrate the importance of

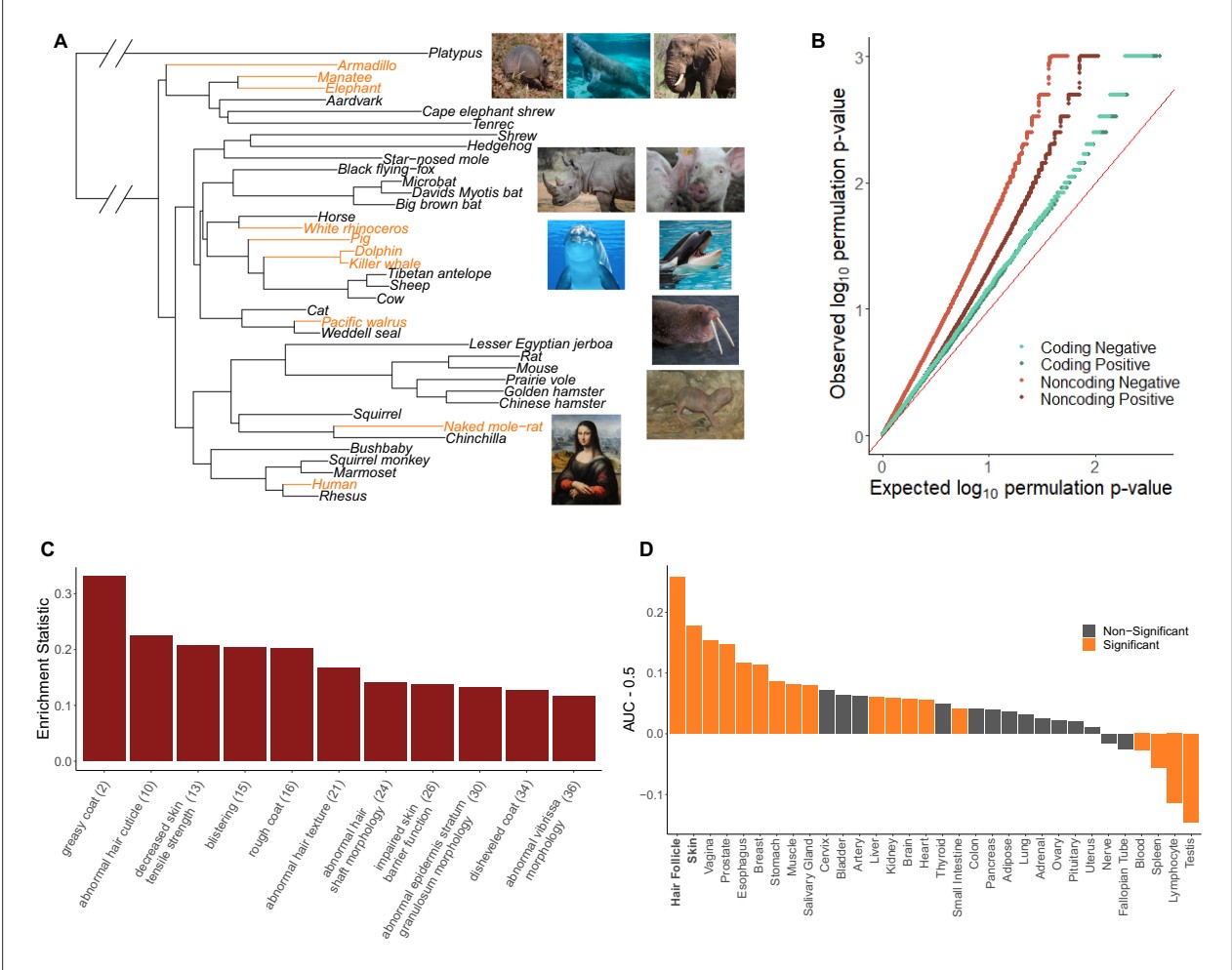

**Figure 1.** Hairless species show an enrichment of hair-related genes and noncoding elements whose evolutionary rates are significantly associated with phenotype evolution. (**A**) Phylogenetic tree showing a subset of the 62 mammal species used for analyses. Note that all 62 species were included in analyses and only a subset are shown here for visualization purposes. Foreground branches representing the hairless phenotype are depicted in orange alongside photographs of the species. (**B**) Q-Q plots for uniformity of permulation p-values for association tests per genetic element for coding and noncoding elements. Shown are both positive associations that indicate accelerated evolution in hairless species and negative associations that indicate decelerated evolution in hairless species. The deviation from the red line (the identity) indicates an enrichment of low permulation p-values – there are more significant permulation p-values than we would observe under the uniform null expectation. This indicates significant evolutionary rate shifts for many genes and noncoding elements in hairless mammals. (**C**) Hair-related Mouse Genome Informatics (MGI) category genes are under significantly accelerated evolution in hairless species. Shown are the AUC (Area Under the Receiver Operating Characteristic curve) values minus 0.5 (maximum enrichment statistic = 0.5, minimum enrichment statistic = –0.5; statistic = 0 indicates no enrichment) for each hair- or skin-related pathway with a permulation p-value≤0.01. In parentheses are the statistic-based ranks of those pathways among all pathways under accelerated evolution in hairless mammals with permulation p-values≤0.01. (**D**) Skin- and hair-expressed genes are under significant evolutionary rate acceleration in hairless species. All genesets except hair follicle are from the GTEx tissue expression database. Hair follicle genes are the top 69 most highly expressed genes from *Zhang et al., 2017* hair follicle RNA sequencing that are not ubiquitously expressed across GTEx tissue types.

The online version of this article includes the following source data and figure supplement(s) for figure 1:

**Source data 1.** Phenotypes.

**Source data 2.** Gene results.

**Source data 3.** Conserved noncoding element results.

**Source data 4.** Positive selection results.

**Source data 5.** Pathway enrichment results.

**Source data 6.** Pathway enrichment results with no KRT or KRTAP genes.

**Source data 7.** hg19 coordinates for conserved noncoding elements.

*Figure 1 continued on next page*

**Source data 8.** mm10 coordinates for conserved noncoding elements.

**Source data 9.** Shared genes in Mouse Genome Informatics (MGI) and tissue expression pathway annotations.

**Figure supplement 1.** Skin-related genes evolve faster in hairless species.

**Figure supplement 2.** Network of pathway annotations shown in *Figure 1C*.

**Figure supplement 3.** Network of pathway annotations shown in *Figure 1D*.

convergent evolution in our analyses. Indeed, the strength of enrichment for hair follicle-related genes among top hits steadily increases as more hairless species share rate shifts in those genes, an indicator of the power of the convergent signal (*Figure 3*). Further, analyses on single species alone only show enrichment for hair follicle-related genes among top hits in 2 hairless species out of 10 – armadillo and pig (*Figure 3—figure supplement 1*). Together, these results demonstrate the importance of testing for convergent evolutionary rate shifts across all hairless mammals to best detect hair-related elements.

Also of important note is that every individual hairless species has thousands of genes with significant rate shifts in that species (*Figure 3—source data 1*). It is impossible to tell which of those rate shifts is associated with hairlessness specifically because the species have many unique phenotypes other than hairlessness that could be responsible for rate shifts in their respective genes. Convergent analyses allow for more concrete identification of hair-related elements by weeding out rate shifts that are not shared across species with the convergent hairless phenotype.

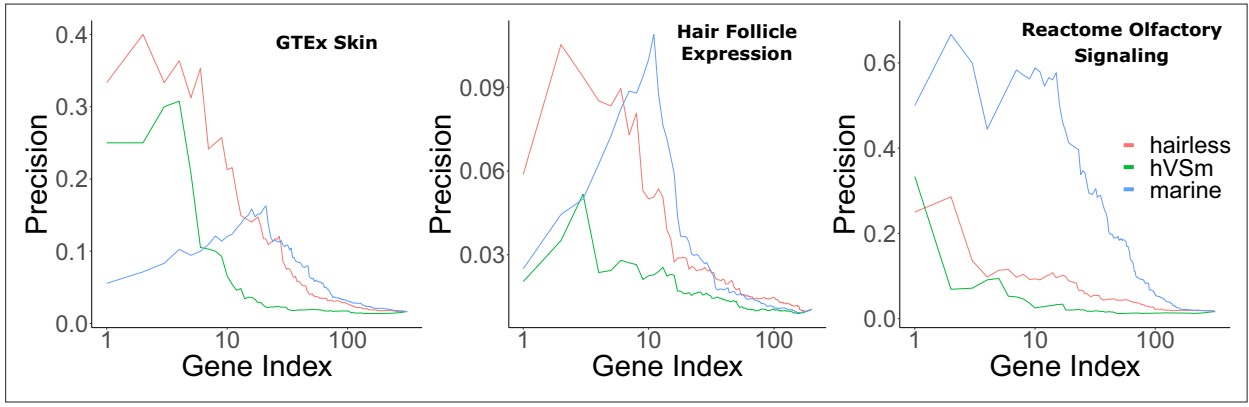

**Figure 2.** Bayes factors reveal the proportion of signal driven by the marine phenotype versus the hairless phenotype. Depicted are precision-recall curves demonstrating how Bayes factors of the contrasting hairless and marine phenotypes rank genes related to skin, hair, and olfaction. Also plotted is a ranking based on the ratio of hairlessness and marine Bayes factors (hVSm = hairlessness Bayes factor/marine Bayes factor). The ratio of the Bayes factors quantifies the amount of support for the hairless phenotype beyond the support for the marine phenotype per gene. In other words, a high Bayes factor ratio indicates a signal of evolutionary convergence associated with hairlessness that is not only driven by signals of convergence in hairless marine mammals. The hairless phenotype had much greater power to enrich for genes expressed in skin (GTEx data) compared to the marine phenotype, indicating that accelerated evolution is driven more strongly by hairlessness. Both the marine and hairless phenotypes enriched for genes in hair follicle expression genes, indicating that both contribute to accelerated evolution of those genes. Olfactory genes, on the other hand, are expected to show acceleration only related to the marine phenotype. As expected, the marine phenotype is much more strongly enriched for olfactory genes than the hairless phenotype.

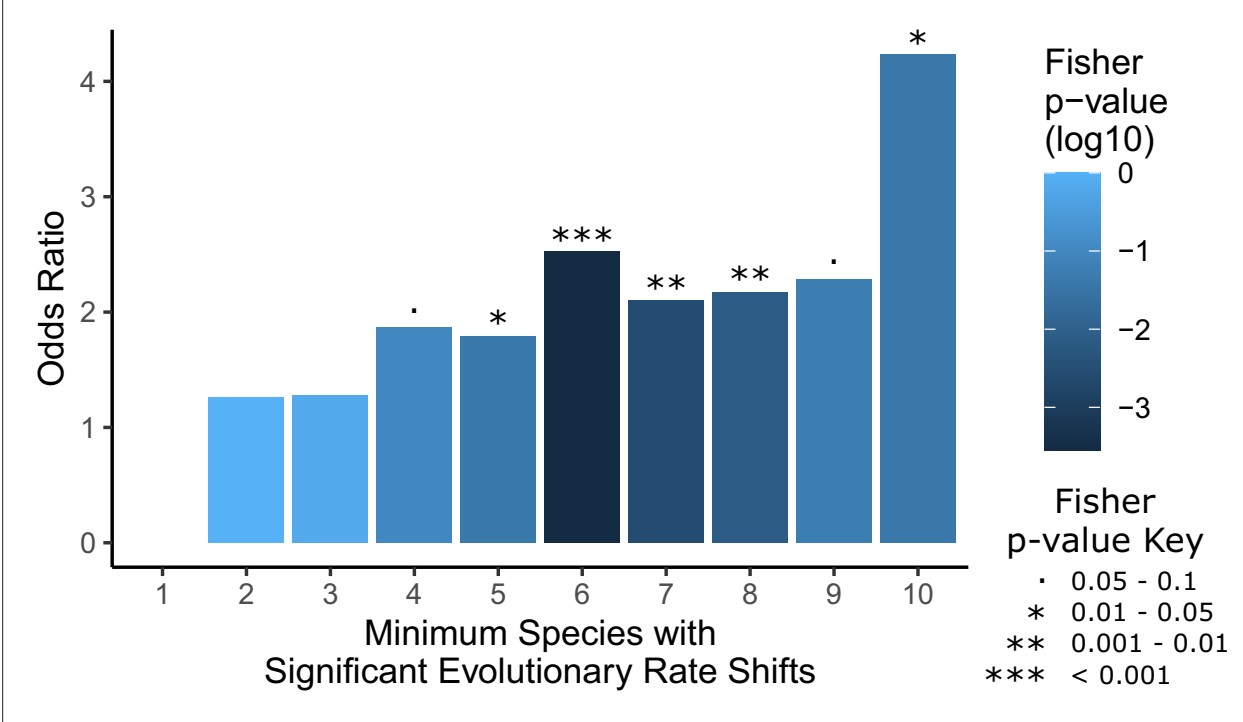

**Figure 3.** Convergent analyses show stronger enrichment for hair-related genes than single-species analyses. Each hairless species was individually tested for a significant rate shift compared to non-hairless species using a Wilcoxon signed-rank test. Then a Fisher's exact test was used to test for an enrichment of hair follicle genes (as shown in *Figure 1D*) with a minimum number of hairless species, ranging from 1 species to all 10 species, with significant rate shifts. Note that the odds ratio for an enrichment with a minimum of one species is not well defined because most genes genome-wide have at least one hairless species with a significant rate shift (18,582 genes out of 18,822 that could be tested), including all hair follicle genes, and their enrichment was not significant (p=0.64). Overall, enrichment strength increases moving from left to right on the plot as the geneset of interest becomes restricted to genes with a larger number of species with rate shifts, although p-values are less extreme because there are simply fewer genes in those categories with higher numbers of species. This demonstrates the convergent signal that allows for detection of hair-related genetic elements based on shared rate shifts.

The online version of this article includes the following source data and figure supplement(s) for figure 3:

**Source data 1.** Species-specific analysis results.

**Figure supplement 1.** Single-species analyses show inconsistent enrichment of hair follicle genes.

### Known hair-related genetic elements evolve faster in hairless species

We used RERconverge to identify genes and noncoding elements evolving at significantly faster or slower rates in hairless species compared to haired species (see 'Methods'). Briefly, the evolutionary rates of genetic elements were compared in hairless versus haired species using a rank-based hypothesis test, and we generated p-values empirically with a newly developed method, termed permulations, that uses phylogenetically constrained phenotype permutations (*Saputra et al., 2020*). The permulation method compares the correlation statistics from the true phenotype to correlation statistics that arise from randomized phenotypes that preserve the relative species relationships. Thus, small p-values indicate a specific association with the hairless phenotype.

We find that quantile–quantile (Q-Q) plots of permulation p-values from hypothesis tests for all genetic elements indicate a large deviation from the expected uniform distribution and thus an enrichment of significant permulation p-values (*Figure 1B*, *Figure 1—source data 2 and 3*). Interestingly, noncoding regions appeared to show even stronger deviation from uniformity than coding regions, perhaps because regulatory changes more strongly underlie the convergent evolution of hairlessness. For both coding and noncoding regions, we show enrichment of significant p-values for both positive and negative evolutionary rate shifts, and the direction of the rate shifts is critical to interpretation. Positive rate shifts imply rate acceleration, which we interpret as a relaxation of evolutionary constraint. While positive rate shifts could theoretically be driven by positive selection,

**Table 1.** Genes whose evolutionary rates are significantly associated with the hairless phenotype with significant parametric p-values, significant permulation p-values, positive statistic, and hairless versus marine Bayes factors (BF) greater than five.

BF marine and BF hairless are BF for those phenotypes individually, while BF hairless/BF marine is the ratio of the two. The ratio of the BF quantifies the amount of support for the hairless phenotype beyond the support for the marine phenotype per gene. In other words, a high BF ratio indicates a signal of evolutionary convergence associated with hairlessness that is not only driven by signals of convergence in hairless marine mammals. Also shown are enrichment statistics for noncoding regions near top genes. Adjusted p-values are Benjamini–Hochberg corrected. Note that permulation p-values observed as 0 were adjusted to 0.001 (the smallest observable permulation p-value) prior to multiple hypothesis testing correction. Cells with missing values (for 'Enrichment statistic (noncoding)' and 'Enrichment p-adj (noncoding)') do not have enough observations to calculate enrichment statistics because too few conserved noncoding elements were detected in the vicinity of those genes. Pseudogene calls are based on premature stop codons reported in *Meyer et al., 2018*.

| Gene | Statistic (gene) | p-adj (gene) | BF hairless/ BF marine (gene) | BF hairless (gene) | BF marine (gene) | Perm p-adj (gene) | Pseudogene (hairless species) | Enrichment statistic (noncoding) | Enrichment p-adj (noncoding) |
|---|---|---|---|---|---|---|---|---|---|
| *FGF11* | 0.403 | 0.205 | 116.4 | 6354.7 | 54.6 | 0.201 | Dolphin | –0.115 | 0.051 |
| *GLRA4* | 0.332 | 0.179 | 22.6 | 1908.3 | 84.3 | 0.201 | Manatee | –0.159 | 0.068 |
| *ANXA11* | 0.328 | 0.179 | 25.5 | 45.2 | 1.8 | 0.201 | No | | |
| *PTPRM* | 0.326 | 0.179 | 51.7 | 4393.6 | 85.0 | 0.201 | No | 0.146 | 1.19e-9 |
| *PKP1* | 0.323 | 0.179 | 5.6 | 2669.0 | 478.9 | 0.201 | No | 0.117 | 0.410 |
| *KRT2* | 0.304 | 0.205 | 2235.7 | 27034.4 | 12.1 | 0.201 | Armadillo, naked mole-rat, orca, manatee | 0.175 | 0.181 |
| *MYH4* | 0.297 | 0.205 | 28.0 | 11447.2 | 409.3 | 0.201 | Dolphin, orca | 0.147 | 0.311 |
| *KRT35* | 0.293 | 0.205 | 8.6 | 1954.5 | 227.3 | 0.201 | Dolphin | 0.142 | 0.211 |

we demonstrate that this is not the case for our top-accelerated genes. Branch-site models to test for positive selection were performed using Phylogenetic Analysis by Maximum Likelihood (PAML) (*Yang, 2007*) on top-accelerated genes. Tests showed little evidence for foreground-specific positive selection; out of 199 genes tested, 27 genes demonstrated hairless acceleration, but all such genes also showed evidence for tree-wide positive selection, suggesting that positive selection was not specific to hairless species although perhaps stronger (*Figure 1—source data 4*). In fact, over half of our top genes from show evidence of pseudogenization, and therefore defunctionalized, in one or more hairless species (*Table 1*; *Meyer et al., 2018*). Thus, regions with positive rate shifts evolve faster in hairless species due to relaxation of evolutionary constraint, perhaps because of reduced functionality driving or in conjunction with the hairlessness phenotype. Negative rate shifts indicate increased evolutionary constraint in hairless species, which implies increased functional importance of a genomic region. While negative shifts are more difficult to interpret in the context of trait loss, they may represent compensatory phenotypic evolution in response to trait loss.

To demonstrate that the statistical signal from individual genes and noncoding regions is meaningful, we evaluated to what extent those RERconverge results enrich for known hair-related elements. We calculated pathway enrichment statistics using a rank-based test and statistics from element-specific results to evaluate whether genes or noncoding elements that are part of a predefined biologically coherent set are enriched in our ranked list of accelerated regions. Using numerous genesets associated with hair growth, such as KRTs, KRTAPs, hair follicle-expressed genes (*Zhang et al., 2017*), skin-expressed genes (*Papatheodorou et al., 2018*), and Gene Ontology (GO) (*Ashburner et al., 2000*), Mouse Genome Informatics (MGI) (*Eppig et al., 2015*), and canonical hair-annotated genes

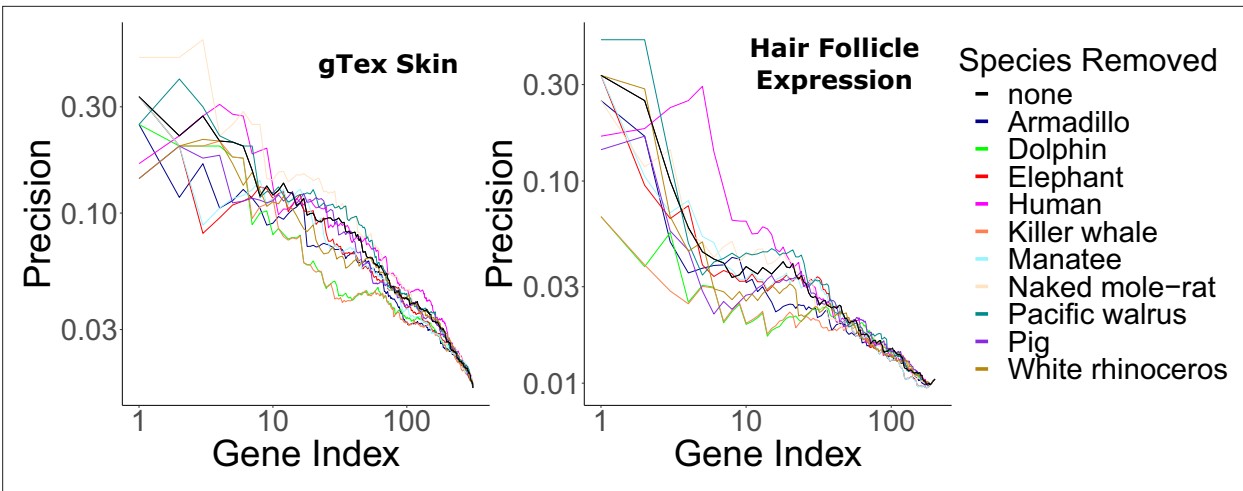

**Figure 4.** Hair-related pathways are enriched for genes with evolutionary rates significantly accelerated in hairless species. Enrichment is consistent even when individual hairless species are removed.

(*Liberzon et al., 2011*), we indeed find that our results are highly enriched for hair-related functions (*Figure 1—source data 5*). As shown in *Figure 1C*, many of the top-enriched MGI phenotypes are hair-related. Likewise, enrichment analyses using the GTEx tissue expression database (*Papatheodorou et al., 2018*) supplemented with hair follicle-expressed genes (*Zhang et al., 2017*) show strong enrichment for both skin and hair follicle genes, as well as signal for other epithelial tissues such as vagina and esophagus (*Figure 1D*). Note that while different MGI and tissue-annotated categories do not contain unique genesets, they are not totally overlapping and tend to cluster into logical higher-order functionalities (*Figure 1—figure supplements 2 and 3*, *Figure 1—source data 9*).

Hair-related pathways remained enriched among rapidly evolving genes even when KRTs and KRTAPs were removed (*Figure 1—source data 6*). This implies that hairless-related genetic changes are not merely structural, but instead they are broadly driven by many genes related to the hair cycle. Similarly, no individual hairless species had an undue impact on enrichment of known hair-related pathways as indicated by consistent findings when individual hairless species were removed from analyses (*Figure 4*).

Investigating a focused list of genesets associated with specific structures of the hair follicle revealed an interesting contrast between coding and noncoding sequence (*Figure 5*). Significantly accelerated genes were primarily within the hair shaft itself for coding sequence. Noncoding regions near genes related to the hair shaft were also under accelerated evolution, and additionally, noncoding regions near genes for the matrix and dermal papilla also showed patterns of decelerated and accelerated evolution, respectively, in hairless species. Since the matrix and dermal papilla play key roles in hair follicle localization, development, and cycling, evolutionary rate shifts in those compartments' noncoding regions suggest that regulatory sequence evolution rather than coding sequence evolution may drive changes in hair follicle formation.

Overall, these results indicate strong enrichment for hair-related function in both protein-coding genes and noncoding regions that are convergently accelerated in hairless species.

## Analyses reveal novel putative hair-related genetic elements

After extensive filtering using RERconverge statistics, Bayes factors, and permulation statistics, several novel putatively hair-related genes were uncovered. As shown in *Table 1* and *Figure 1—figure supplements 4–11*, the top-accelerated gene associated with hairlessness with strong support for hairless-related signal as opposed to marine-related signal was *FGF11*. While *FGF11* has no known role in hair growth, its expression is highly enriched in the skin and other fibroblast growth factor genes are known to be related to hair growth (*Kawano et al., 2005*; *Lee et al., 2019*; *Nakatake et al., 2001*; *Rosenquist and Martin, 1996*; *Suzuki et al., 2000*). Together, these observations support *FGF11* as another strong candidate for hair-related function.

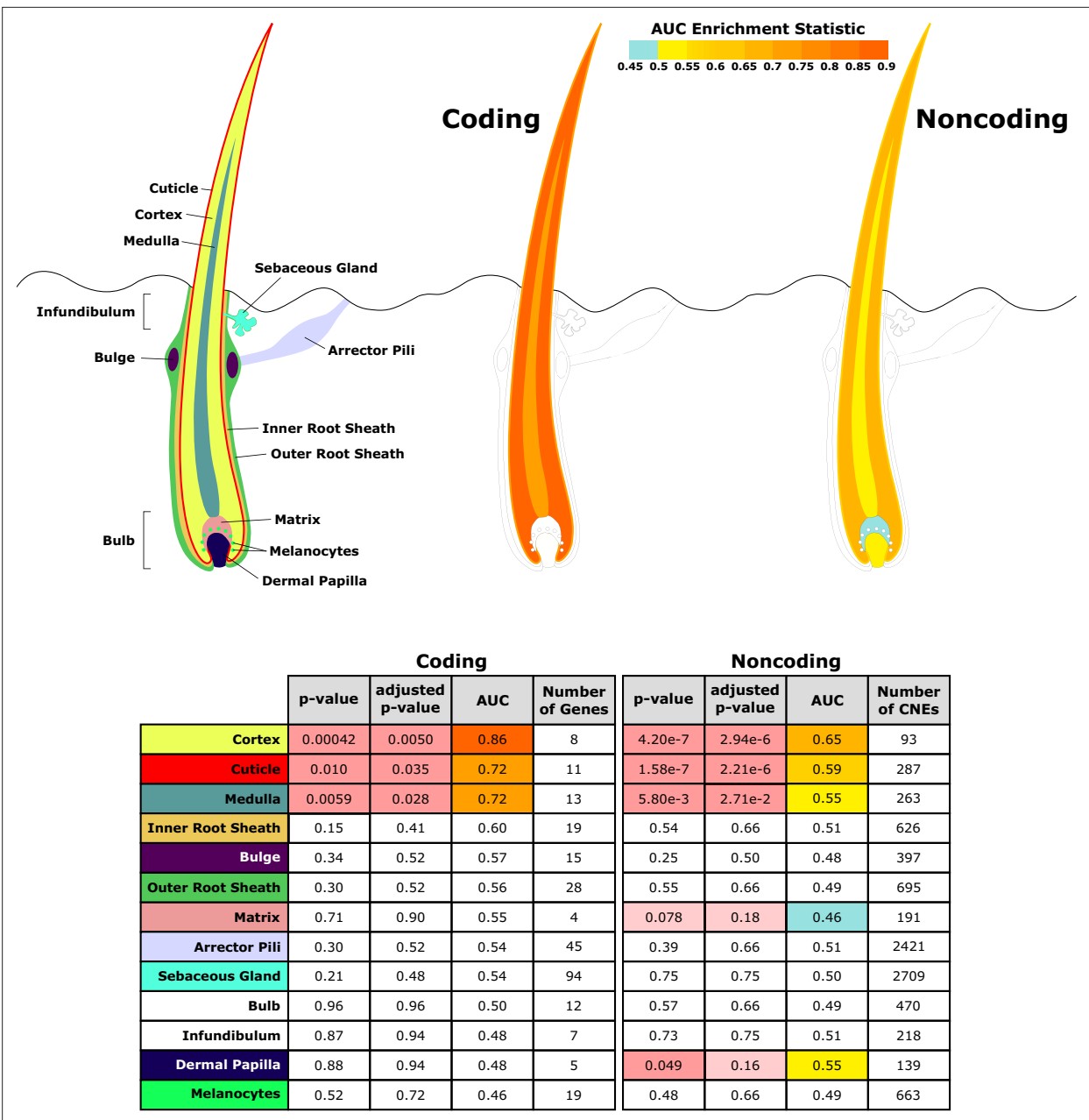

**Figure 5.** Diagram of hair shaft and follicle with shading representing region-specific enrichment for coding and noncoding sequence. Both coding and noncoding sequence demonstrate accelerated evolution of elements related to hair shaft (cortex, cuticle, and medulla). Noncoding regions demonstrate accelerated evolution of matrix and dermal papilla elements not observed in coding sequence. All compartment genesets were compiled from Mouse Genome Informatics (MGI) annotations that contained the name of the compartment except the arrector pili geneset (*Santos et al., 2015*).

The second-ranked gene, *GLRA4*, a glycine receptor subunit, is more difficult to interpret because while generally conserved across mammals, it is a pseudogene in humans, so it has been relatively less studied. Glycine receptors are often involved in motor reflex circuits (*Callister and Graham, 2010*), and thus with respect to any functional relevance to hair we hypothesize that GLRA4 may contribute to regulating the reflexive piloerection response (hairs 'standing on end') observed in many mammals.

Other top-accelerated genes are *KRT2*, *KRT35*, *PKP1*, and *PTPRM*, all of which are known hair-related genes. *KRT2* protein product localizes in the hair follicle and may play a role in hair and skin coloration (*Cui et al., 2016*), and *KRT35* is a known target of *HOXC13* and is essential for hair differentiation (*Lin et al., 2012*). *PKP1* mutations lead to ectodermal dysplasia/skin fragility syndrome, which

includes abnormalities of both skin and hair development (*Sprecher et al., 2004*). *PTPRM* regulates cell–cell communication in keratinocytes (*Peng et al., 2015*).

The remaining accelerated genes are also plausibly connected to skin- and hair-related functions. *ANXA11* has been strongly linked to sarcoidosis in humans (*Hofmann et al., 2008*), an inflammatory disease in epithelial tissue. *MYH4*, a myosin heavy-chain protein, has surprisingly also been implicated in skin and hair growth, both through upregulation during hair follicle cycling and skin healing (*Carrasco et al., 2015*) and upregulation in response to overexpression glucocorticoid receptors that drive hair follicle morphogenesis (*Donet et al., 2008*). Note that in both cases of *MYH4* upregulation it was the only myosin with significantly different expression in the tissues studied, suggesting a unique role for the protein in skin and hair growth.

In addition to identifying genes with significant evolutionary rate shifts in coding sequence, we have also found many other protein-coding genes with significant enrichment of hairless-accelerated noncoding elements in their vicinity (see Dryad and *Figure 6*). There is a global trend in correlation between evolutionary rate shift statistics for protein-coding regions and enrichment statistics for their nearby noncoding regions (Pearson's rho = 0.177). Concordance between accelerated evolutionary rates in genes and their nearby noncoding regions is particularly strong for *KRT*s and *KRTAP*s, which are known to be skin- and hair-related (*Figure 6B*) – out of 69 *KRT*s and *KRTAP*s for which noncoding enrichment could be calculated, 66 showed accelerated evolution in both protein-coding sequence and noncoding regions. However, across all genes with strong signals for nearby noncoding regions under accelerated evolution in hairless mammals (permulation p-value≤0.03), acceleration in the coding sequence itself spans a wide range of values (*Figure 6A*), and in many cases there is little evidence of evolutionary rate shifts in the coding sequence. This range likely reflects the requirement that some protein-coding sequences remain under strong evolutionary constraint because of their continued importance in non-hair-related tissues.

Top-ranked genes with accelerated nearby noncoding regions include several known hair-related regulator genes (*ELF3*, *FOXC1*, and others) (*Figure 6C*). *FOXC1* is a transcription factor involved in maintaining the hair follicle stem cell niche (*Lay et al., 2016*; *Wang et al., 2016*) and *ELF3* is known to regulate transcription of keratin genes (*Aldinger et al., 2009*). These genes showed no coding region acceleration, which is expected since they are highly pleiotropic. Regulatory proteins tend to have many functions – for example, in addition to their hair-related functions, *FOXC1* regulates embryonic development (*Brembeck et al., 2000*; *Seo et al., 2006*) and *ELF3* is involved in the epithelial-to-mesenchymal transition (*Sengez et al., 2019*) – so we expected to observe no loss of constraint in the coding sequence for those proteins. Instead, changes to regions that regulate expression of those regulatory proteins appear to be driving the convergent evolution of hairlessness. While regulation of transcription factor expression is highly complex, our analysis pinpoints regions that are candidates for hair-specific regulation.

The global analysis of noncoding regions also revealed undercharacterized regions (*CCDC162-SOHLH2*, *FAM178B*), and regions that may plausibly be connected to hair or skin (*UVSSA* [*Sarasin, 2012*], *OLFM4* [*Jaks et al., 2008*; *Muñoz et al., 2012*], *ADRA1D* [*Rezza et al., 2016*]). These noncoding regions are excellent candidates for further experimental analyses to explore their role in regulating hair and skin growth, development, and cycling.

Perhaps even more so than genes and their regulatory regions, microRNAs are strong candidates for hair-related functions. A key component of hair follicle cycling is persistence of stem cells, and microRNAs are known to be important players in stem cell regulation (*Peng et al., 2015*). Too small to be analyzed via their sequence alone using our analysis strategy, we mapped noncoding regions to nearby microRNAs and performed enrichment analyses to identify groups of microRNA-associated noncoding regions enriched for significant association with the hairlessness phenotype (*Figure 7A*). The top-enriched microRNA with rapidly evolving nearby noncoding regions was mir205, a microRNA known to be associated with skin and hair development (*Wang et al., 2013*). *Mir205* is readily studied because its host gene (*mir205hg*) is long enough to be captured using standard methods, including bulk RNA sequencing. Reanalyzed data from a previous study (*Zhang et al., 2017*) focused on coding sequence revealed read pileups at *mir205hg* even without microRNA-specific capture methods (*Figure 7C and D*). Through our study, we now know which noncoding regions in the gene desert around *mir205* potentially control its expression in hair follicles as opposed to other tissues. Through this scan for associated noncoding elements, we similarly identified several poorly characterized

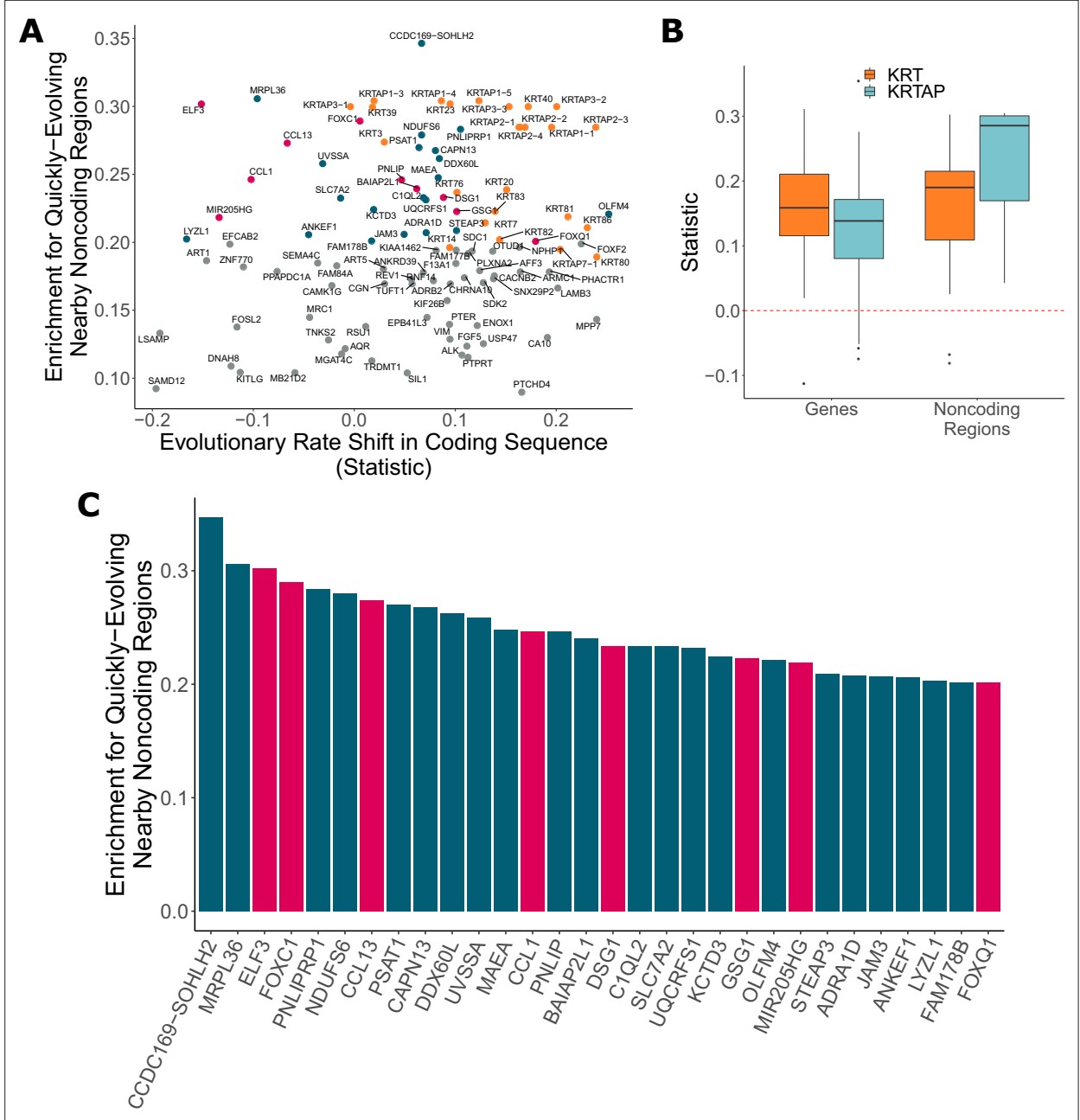

**Figure 6.** Noncoding regions near hair-related genes evolve faster in hairless species. (**A**) Genes with a significant enrichment for quickly evolving nearby noncoding regions (permulation p-value of 0.03 or less) only sometimes demonstrate evolutionary rate shifts in their protein-coding sequences. In orange are keratins and keratin-associated proteins, which tend to show accelerated evolutionary rates in both genes and nearby noncoding regions. In pink are top genes, also in pink in panel (**C**). In blue are all other genes in panel (**C**). (**B**) Keratin (*KRT*) and keratin-associated protein (*KRTAP*) genes and nearby noncoding sequence show enrichment for accelerated evolutionary rates. Shown are rate shift statistics for genes and enrichment statistics for noncoding regions. (**C**) Many top-ranked genes for nearby quickly evolving noncoding regions are hair-related. Depicted are the top 30 genes (*KRT*s and *KRTAP*s excluded) based on enrichment statistic with enrichment permulation p-value of 0.03 or less. No genes had significant evolutionary rate shifts in coding sequence except *OLFM4*, which evolves faster in hairless species. In pink are genes with hair-related functions in the literature (citations: *ELF3* [**Blumenberg, 2013**], *FOXC1* [**Lay et al., 2016**], *CCL13* [**Michel et al., 2017**; **Suárez-Fariñas et al., 2015**], *CCL1* [**Nagao et al., 2012**], *DSG1* [**Zhang et al., 2017**], *GSG1* [**Umeda-Ikawa et al., 2009**], *MIR205HG* [**Wang et al., 2013**], *FOXQ1* [**Ashburner et al., 2000**; **Carbon et al., 2019**]).

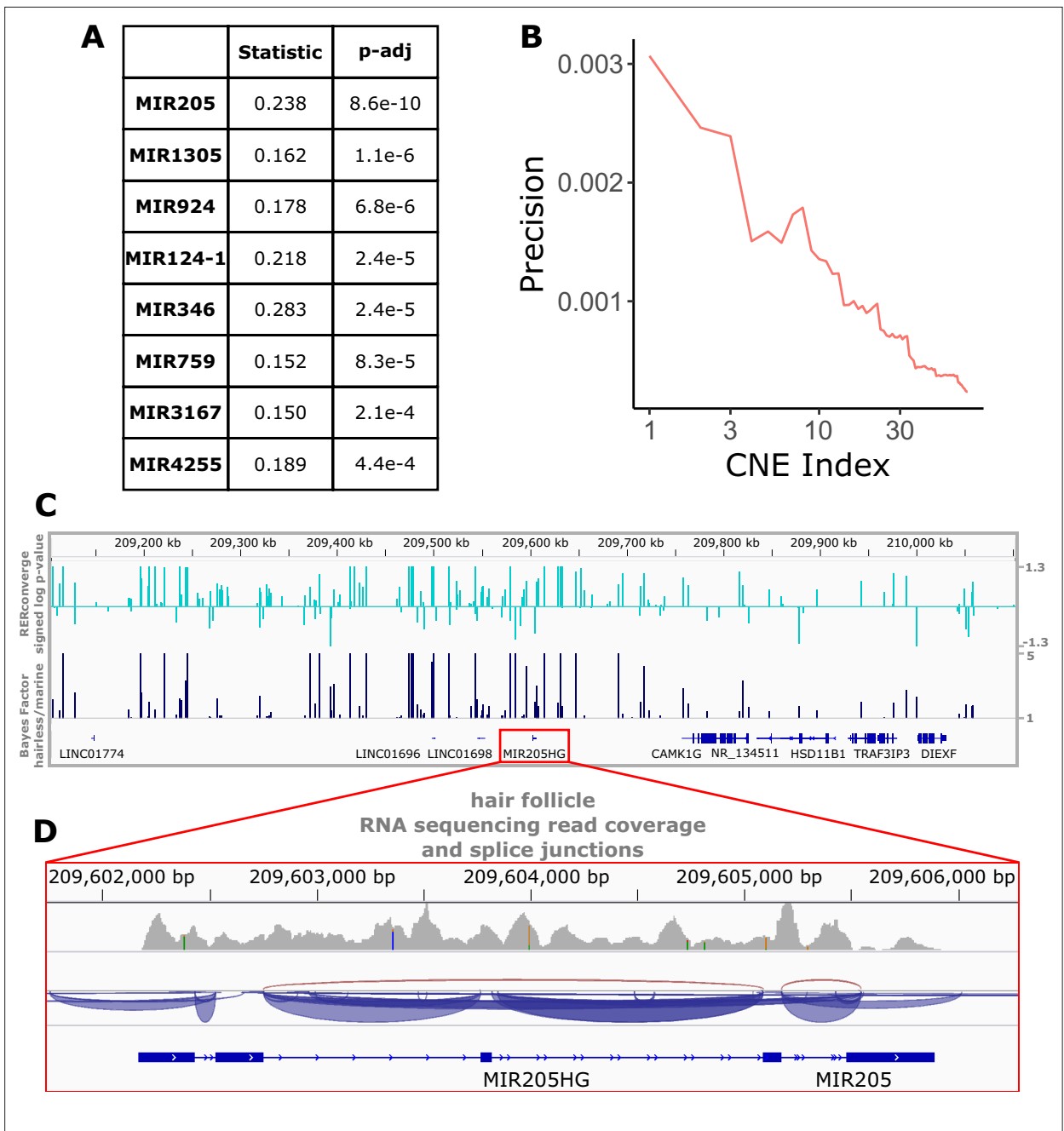

**Figure 7.** Top miRNAs with nearby noncoding regions with evolutionary rates significantly associated with the hairless phenotype. (**A**) Wilcoxon rank-sum enrichment statistics and Benjamini–Hochberg corrected p-values for top-ranked miRNAs. (**B**) Precision recall curve of statistic ranks for CNEs near *mir205* demonstrates an enrichment of CNEs with accelerated evolution near *mir205* compared to all noncoding regions near microRNAs. (**C**) The chromosomal region around *mir205* shows a large number of CNEs accelerated in hairless species, as seen for RERconverge and Bayes factor scores. Note the relative decline of peaks in the vicinity of nearby protein-coding genes such as *CAMK1G* to the right. (**D**) *mir205* is well-known to be associated with hair and skin growth and structure. Its transcriptional unit on chromosome 1 shows clear read pileups from hair follicle RNAseq data (*Zhang et al., 2017*). Gray peaks represent the number of RNAseq read coverage, and blue curves represent splice junctions.

microRNAs with significant hair-related signal that are less studied and are also strong candidates for hair-related functions (*Figure 7A*). Furthermore, we have identified the precise noncoding regions that likely control their expression in the context of hair and hair follicles (*Figure 1—source data 5*).

## Discussion

These analyses successfully used RERconverge, a method to link convergent evolutionary rates of genetic elements with convergent phenotypes, to identify known hair-related genes in mammals. In addition to identifying known genes, other understudied genes and microRNAs were also identified as key plausible targets for further inquiry into the genetic basis of hairlessness, and a suite of putative regulatory elements associated with hair and skin were uncovered.

The top-ranked gene was *FGF11*, a fibroblast growth factor gene. It evolved faster in hairless species due to relaxation of evolutionary constraint, indicating that it has reduced functionality in hairless species. Fibroblast growth factors are readily studied for a variety of functions, but the precise functionality of *FGF11* is unknown. The gene may be associated with cancer development through interaction with T-cells (*Ye et al., 2016*), and it has also been implicated in tooth development in mice (*Kettunen et al., 2011*). Interestingly, the gene related to hairlessness in Mexican hairless dogs is also related to dentition (*Drögemüller et al., 2008*), implying plausibility for a hair-related gene to also be tooth-related. Furthermore, numerous fibroblast growth factor genes have been studied in relation to hair growth (*Rosenquist and Martin, 1996*), including work that found errors in *FGF5* resulted in longer hair in goats (*Li et al., 2019*), *FGF5* and *FGF7* regulation controlled hair anagen phase in mice (*Lee et al., 2019*), and *FGF2* stimulated hair growth when applied to mouse skin (*Xu et al., 2018*). *FGF11* is an excellent candidate to perform similar tests for hair-related functions, among other high-scoring genes in the list such as *MYH4* and *ANXA11*.

Compared to coding sequence, study of noncoding regions is more challenging for several reasons. First, identifying such regions genome-wide is difficult because they lack the defining characteristics that genes share, such as start and stop codons, and thus finding putative regulatory elements using sequence alone is an ongoing area of study. Our strategy of using conserved regions as putative regulatory elements likely misses many real regulatory sequences while simultaneously capturing conserved elements with no regulatory function. However, our method of sequence selection is also unbiased and provides a robust set of sequences to analyze, many of which likely do have regulatory functions. Second, validating our findings from noncoding regions is difficult because few CNEs have known functions. Therefore, to validate our noncoding results, we mapped noncoding regions to nearby genes and inferred CNE functions based on the functions of those genes. Such proximity-based mapping has known flaws because enhancers can have distal effects and chromatin state controls enhancers' access to genes regardless of distance. However, despite all of the potential sources of error, we identify global signal for noncoding regions under accelerated evolution in hairless species (*Figure 1B*) and signal for hair-related acceleration of noncoding regions (*Figures 5–7*).

Further analyses of noncoding regions revealed an interesting deviation from signals of accelerated evolution in coding regions. Namely, coding regions primarily showed acceleration in genes related to texture and the structure of the hair shaft itself. Noncoding regions, on the other hand, showed accelerated evolution near genes related to the dermal papilla and the matrix. Both regions are essential for hair growth. The dermal papilla is the master controller of hair follicle development and hair growth, and it has in fact been repeatedly shown to be sufficient to cause hair growth. Dermal papilla cells, when transplanted to hairless skin such as footpads, have consistently been shown to result in development of hair follicles (*Jahoda et al., 1984*; *Jahoda et al., 1993*; *Reynolds and Jahoda, 1992*). Since all mammals are capable of growing hair and do have at least some hair at some point in their life cycles, these findings imply that function of genes related to the dermal papilla must be preserved, and spatial and temporal changes in hair growth may be driven by noncoding regions. Much like the dermal papilla, the hair follicle matrix is essential for hair growth – mitotically active matrix cells give rise to all other inner hair structures, including the hair shaft and the root sheath. Early-stage matrix differentiation can even progress without dermal papilla signaling (*Mesler et al., 2017*). Hair cannot exist without the dermal papilla and matrix, and alterations to their related noncoding regions could plausibly have a large impact on hair growth capabilities. Changes to their associated regulatory regions, on the other hand, may be more flexible and allow for the changes in hair localization, texture, and density that we observe in near-hairless mammals.

Other genes with nearby accelerated noncoding regions likewise demonstrate conservation in protein-coding sequence, possibly because of strong pleiotropy of hair- and skin-related genes. In fact, among the top-ranked non-keratin genes with quickly evolving nearby noncoding regions, only one gene showed a significant evolutionary rate shift in protein-coding sequence (*Figure 6*). *FOXC1* and *ELF3*, among the top-ranked genes, are strongly linked to hair and skin development (*Brembeck et al., 2000*; *Lay et al., 2016*; *Wang et al., 2016*) but also have other essential functions (*Aldinger et al., 2009*; *Sengez et al., 2019*; *Seo et al., 2006*). Our findings imply that many hair-related genes may have similar pleiotropy preventing accelerated evolution of coding sequence in hairless species. Instead, plasticity of gene regulation through accelerated evolution of noncoding regions may allow for the evolution of hairlessness.

Additionally based on noncoding sequences near regions of interest, mir205 was found to be the top-ranked microRNA with nearby noncoding sequences under accelerated evolution. Mir205 is well-established microRNA related to hair and skin development (*Wang et al., 2013*), and it thereby serves as a strong validation that signals of convergent evolution are successfully identifying hair-related elements. The second-ranked mir1305 has been implicated in skin functionality with significantly different expression levels in damaged versus healthy skin (*Liang et al., 2012*). Numerous microRNAs have been implicated in hair- and skin-related functions (*Andl and Botchkareva, 2015*; *Fu et al., 2014*), likely a subset of the total elements involved in hair growth. In general, microRNAs are likely key players in hair follicle cycling because of their importance in stem cell regulation (*Peng et al., 2015*), and the microRNAs and their associated noncoding regions identified in this work serve as a valuable list of candidates for further inquiry. Likewise, noncoding regions near other hair-related genes are also under accelerated evolution in hairless species and may regulate hair- and skin-related functions. Further, some undercharacterized and plausibly hair- and skin-related genes, such as *CCDC169-SOHLH2* and *FAM178B*, have nearby accelerated noncoding regions and thus identify those genes and their regulatory regions as candidates for further experimental testing.

This study has revealed a slew of fresh candidate genes, noncoding regions, and microRNAs putatively associated with hair growth. Notably, it avoids identifying species-specific genetic changes that could be associated with any number of phenotypes unique to each species and instead looks for general hair-related genomic elements relevant across many species. As a genome-wide scan across a large swath of the mammalian phylogeny, it represents not only a step toward fully understanding hair growth, but also understanding the evolution of hair across all mammals.

## Methods

### Calculating body size-regressed relative evolutionary rates

The RERconverge package in R was used to generate phylogenetic trees for each gene and noncoding region in which branch length represented the amount of evolutionary change, or the number of nonsynonymous substitutions, that occurred along that branch as described in several previous publications (*Kowalczyk et al., 2019*; *Kowalczyk et al., 2020*; *Partha et al., 2019*). Alignments for 19,149 genes in 62 mammal species were obtained from the UCSC 100-way alignment (*Blanchette et al., 2004*; *Harris, 2007*; *Kent et al., 2002*). The topology used to generate element-specific trees is included below under 'Phylogenetic trees'.

Likewise, alignments for 343,598 conserved noncoding elements were extracted based on phastCons conservations scores across the 62 mammal species and the blind mole-rat (*Nannospalax galili*) (*Siepel et al., 2005*). Briefly, the full set of conserved elements across 46 placental mammals and their respective phastCons scores were downloaded from the UCSC genome browser (*Kent et al., 2002*) from the hg19 (human genome) 'Cons 46-way' track (phastConsElements46wayPlacental). Regions that overlapped coding regions were removed using the UCSC genome browser 'Intersection' utility and the 'Genes and Gene Predictions' annotations from the 'GENCODE V28lift37' track. Elements with phastCons scores greater than 350 were maintained, and elements less than 10 base pairs apart were merged. Finally, elements with fewer than 40 base pairs were discarded to result in the final 343,598 regions. Orthologs for all 62 mammals were downloaded from the UCSC 100-way alignment. Blind mole-rat elements were added based on the pairwise alignment between hg38 (human genome) and *N. galili* genome (*Zerbino et al., 2015*) by first mapping hg19 coordinates to hg38 coordinates

(*Figure 1—source data 7 and 8*). Orthologs were added to the 62 mammal species alignments using MUSCLE (*Edgar, 2004*).

Alignments were used to generate evolutionary rate trees based on a well-established topology of the mammalian phylogeny in the PAML program (*Meyer et al., 2018*; *Yang, 2007*). Briefly, RERconverge was used to convert evolutionary rate information from each gene- or noncoding element-specific tree by correcting for the mean–variance relationship among branch lengths and normalizing each branch for the average evolutionary rate along that branch such that the final branch length was relative to the expectation for that branch (*Partha et al., 2019*).

The resulting relative evolutionary rates were used to calculate body size-regressed relative evolutionary rates. Using adult weight information for the 62 mammal species obtained from the Anage Animal Aging and Longevity Database (*Tacutu et al., 2018*), RERconverge functions were used to predict body size phenotype values throughout the mammalian phylogeny. Residuals from a linear model fitted to the phenotype values and the relative evolutionary rates for each gene and conserved noncoding element were extracted and used as the body size-regressed relative evolutionary rates for that element.

RER matrices and phylogenetic trees are available on Dryad.

## Defining hairless species

Since all mammals have some hair during at least one stage of life, no species are truly hairless. Therefore, classification of species as 'hairless' versus 'haired' was qualitatively based on density of hair covering and quantitatively based on the impact of removing species on the hair-related signal detected during analyses. Tendency was to err on the side of leniency when assigning species as hairless – any species with reduced hair quantity was classified as hairless.

Extant species classified as hairless were armadillo, elephant, white rhinoceros, pig, naked mole-rat, human, and marine mammals (manatee, pacific walrus, dolphin, and orca). The hairless set comprised all but one marine mammal in the 62 mammal species (the furry Weddell seal is not included in the hairless set). The only nonextant species classified as hairless was the orca-dolphin ancestor (the cetacean ancestor) because that species was likely also a hairless marine species (*Chen et al., 2013*; *Nery et al., 2014*). The elephant–manatee ancestor was not classified as hairless because modern elephants have known extinct hairy sister species (wooly mammoths) that diverged after the elephant–manatee divergence (*Roca et al., 2009*). Thus, classifying the elephant–manatee ancestor as hairy was the most parsimonious phenotype assignment for the afrotherian clade. The classification was also supported by the data, which indicated a stronger signal for skin-related genes when the elephant–manatee ancestor was classified as hairy (*Figure 1—figure supplement 1*).

Although some species are undeniably hairy (dog, cat, sheep, etc.) and some are undeniably relatively hairless (orca, dolphin, elephant, etc.), some species are borderline cases. For example, the tenrec and hedgehog appear to have 'spikes' rather than hair. However, tenrec and hedgehog spikes (as well as porcupine quills) are modified hairs (*Leon Augustus, 1920*), so we classified tenrec and hedgehog as hairy. Armadillo, pig, and human are likewise classified as hairless species but have relatively greater hair quantity than the other hairless species. The armadillo, like the tenrec and hedgehog, has a unique external modification, but unlike the tenrec and hedgehog, the armadillo's shell is made of bone, not hair (*Chen et al., 2011*), so we classified the armadillo as hairless. Pig and human, on the other hand, have nonmodified skin that is nearly completely covered in hair (and in the case of humans, the hair is quite dense in some body areas), but both species have large swaths of body area where hair is so sparse that sun-exposed skin is clearly visible. Both species were classified as hairless due to this pervasive low hair density. To assess the impact of species assignment on skin- and hair-related signal, hairless species were systematically removed and relevant enrichment statistics were recalculated. No specific species has a consistently detrimental impact on enrichment for genesets of interest (*Figure 4*).

## Calculating element-specific association statistics

For each genetic element, evolutionary rates for haired species versus hairless species were compared using Kendall's tau. Haired species included ancestral species inferred to be haired in addition to extant haired species. Resulting p-values were multiple hypothesis testing corrected using a standard Benjamini–Hochberg correction (*Benjamini and Hochberg, 1995*).

In addition to calculating parametric p-values, empirical p-values were calculated using a novel permulation strategy modified from a similar strategy developed for continuous phenotypes (*Kowalczyk et al., 2020*). First, 1000 null phenotypes were generated by using Brownian motion phylogenetic simulations and assigning the top 10 values as hairless species. Resulting phenotypes were backpropagated along the phylogeny to ensure that final null phenotypes contained a total of 11 foreground species with only a single ancestral species classified as hairless. Such a procedure matched the organization of null phenotype values to true phenotype values. Hypothesis testing was repeated using all null phenotypes, and the empirical p-values were calculated as the proportion of permulations with statistics as extreme or more extreme than the parametric statistic for the real phenotype values.

## Calculating element-specific Bayes factors

In addition to calculating element-specific association statistics, Bayes factors were calculated for each gene using the marine and hairless phenotypes using the BayesFactor R package (*Morey and Rouder, 2021*). These values were calculated to disentangle the two phenotypes, which are heavily confounded since nearly all marine mammals in the genome alignment used for this work are hairless.

Briefly, Bayes factors are a Bayesian approach complementary to more standard statistical tests. Instead of returning statistics and p-values, Bayes factors directly quantify the amount of support for an alternative hypothesis. For example, a Bayes factor value of 5 for a particular statistical test would indicate five times more support for the alternative hypothesis than the null hypothesis. Bayes factors can also be used to compare different alternative hypotheses by calculating the ratio of two Bayes factors.

When considering the hairless phenotype, we use Bayes factors to quantify the support for a linear model predicting phenotype using evolutionary rate information from each gene, with a higher Bayes factor indicating greater support. We perform this calculation for two alternative hypotheses: (1) a gene shows different evolutionary rates in hairless versus hairy species, and (2) a gene shows different evolutionary rates in marine species versus nonmarine species. The ratio of Bayes factors between the hairless and marine phenotypes quantifies the level of support of one phenotype over the other and thus can be used to tease apart intricacies of the two heavily confounded phenotype. When the Bayes factor for the hairless phenotype is much larger than the Bayes factor for the marine phenotype, that indicates stronger support for signal driven by hairlessness.

## Calculating enrichment statistics

Enrichment statistics were calculated using MGI genesets (*Blake et al., 2003*), GTEx tissue annotations (*Papatheodorou et al., 2018*), GO annotations (*Ashburner et al., 2000*; *Carbon et al., 2019*), and genes highly expressed in hair follicles (*Zhang et al., 2017*). The 70 hair follicle-specific genes were obtained by selecting the top 200 hair follicle-expressed genes and removing genes that were included in the top 10,000 genes with the highest minimum median expression across GTEx tissues, that is, ubiquitously expressed genes. Noncoding regions were mapped to annotations via distance from relevant genes – regions within 10,000 bases of a gene were assigned to that gene and its pathways. Noncoding regions were also mapped to microRNA coordinates using the same distance-based metric. All annotations are available on Dryad.

Pathway enrichment statistics were calculated using the Wilcoxon rank-sum test, which compares ranks of foreground values for elements in a pathway to background values for nonpathway elements. For each gene or noncoding element, the sign of the statistic times the log of the p-value were used to generate ranks. Empirical p-values from permulations were also generated using the same null phenotypes used for individual elements and detailed in previous work (*Kowalczyk et al., 2020*).

## Permulations

In addition to computing parametric statistics directly from standard statistical tests, empirical p-values were also calculated using a permulation strategy. Permulations were used to generate null phenotype values, and the empirical p-value was calculated as the proportion of null statistics as extreme or more extreme than the observed parametric statistics. Such a strategy corrects for a nonuniform empirical null distribution at the gene level (*Figure 1*) and nonindependence among genetic elements at the pathway level (*Saputra et al., 2020*).

## Positive selection tests

For top-ranked genes under accelerated evolution in hairless species, all KRT and KRTAP genes, and various genes in top-ranked pathways under accelerated evolution in hairless species, branch-site models to test for positive selection were performed to identify whether rapidly evolving genes were undergoing positive selection or merely under relaxation of constraint. Such models were performed using a subset of the full 62 species mammalian phylogeny as shown in the 'Phylogenetic trees' section below.

Significance of relaxation of constraint for hairless species was assessed using likelihood ratio tests (LRTs) between branch-site neutral (BS Neutral) and its nested null model M1 (sites neutral model) in PAML (*Yang, 2007*). Similarly, LRTs between branch-site selection model (BS Alt Mod) and its null BS Neutral were used to infer positive selection in hairless species. For each test, p-values were estimated using the chi-square distribution with 1 degree of freedom. Phylogeny-wide relaxation of constraint was additionally quantified using the LRTs between M2 (sites selection model) vs. M1 (sites neutral model) and M8 (sites selection model) vs. M8A (sites neutral model), respectively. Prior to performing the mammal-wide tests, hairless foreground species were removed to estimate significance of relaxation of constraint and positive selection from only the background mammalian branches. Removing hairless species allowed us to distinguish whether genes were under positive selection in all mammal species or only in hairless mammal species. Genes with significant signals of positive selection and nonsignificant signals of phylogeny-wide acceleration were inferred to be under positive selection (*Kowalczyk et al., 2021*).

## Species-specific analyses

Further tests to identify genomic elements with evolutionary rate shifts in individual hairless species were conducted as follows. First, RERs from all extant species across all genes were retained and RERs for ancestral species were discarded. Next, RERs for hairless species were removed. Finally, for each gene, a Wilcoxon signed-rank test was run to individually compare the RER of each hairless species to the RERs for the non-hairless species.

Results of the Wilcoxon signed-rank test were pooled in two different ways. First, for each gene, the number of species with significant rate shifts was counted. Using those counts, a Fisher's exact test was used to calculate a hair gene enrichment statistic between genes with cutoffs ranging from 1 to 10 species representing the minimum number of species with significant rate shifts. Hair follicle annotations used for enrichment are available on Dryad.

A second set of enrichment statistics was calculated using the same annotations to test for an enrichment of hair genes with significant rate shifts in each hairless species individually. To complement these analyses, a second set of non-hairless species was selected and tested using the same procedure as the hairless species. In short, those species were removed from the set of all RERs and then each was individually compared to the background set of RERs from species not in the set using a Wilcoxon signed-rank test.

## Phylogenetic trees

### Master tree topology with average branch lengths

((((((((((((((ailMel1:0.03854019703,((lepWed1:0.02002160645,odoRosDi:0.02064385875):0.01734764946,musFur1:0.04613997497):0.002879093616):0.009005888384,canFam3:0.05339127565):0.01185166857,felCat5:0.05020331605):0.03285617057,((((((bosTau7:0.02168740723,((capHir1:0.01157093136,oviAri3:0.01246322594):0.0049716126,panHod1:0.01522587482):0.01465511149):0.0662523666,(orcOrc1:0.006371664911,turTru2:0.01086552617):0.06014682602):0.01216198069,susScr3:0.0796745271):0.006785823323,(camFer1:0.01240650215,vicPac2:0.01096629635):0.06374554586):0.02551888691,(cerSim1:0.04977357056,equCab2:0.061454379):0.02510111297):0.0033121468 6,((eptFus1:0.03248546656,(myoDav1:0.02344332842,myoLuc2:0.01567729315):0.02193849809):0.09455328094,(pteAle1:0.005833353548,pteVam1:0.01611220178):0.07567400302):0.02385546003):0.002057771224):0.004845253848,(conCri1:0.1239823369,(eriEur2:0.1696142244,sorAra2:0.1934205791):0.02079474546):0.0235875333):0.01477733374,((((((chrAsi1:0.1017903453,echTel2:0.1749615473):0.01592632003,eleEdw1:0.1516860647):0.006610995228,oryAfe1:0.08326528894):0.008243787904,(loxAfr3:0.06812658238,triMan1:0.06198982615):0.0224994529):0.03384011363,dasNov3:0.1342602666):0.005989703247,(((macEug2:0.1270943532,sarHar1:0.09944141622):0.02717055

443,monDom5:0.1181200712):0.1802966572,ornAna1:0.4322118716):0.2206952867):0.0131643
6193):0.01425600689,((((((cavPor3:0.09048639907,(chiLan1:0.05332953299,octDeg1:0.08476954109
):0.01287861561):0.02118937782,hetGla2:0.08588673524):0.07432515556,speTri2:0.0889642464
2):0.006291577528,((((criGri1:0.04084640027,mesAur1:0.04456203524):0.02314125062,micOch1
:0.06932402649):0.01947113467,(mm10:0.05273642272,rn5:0.05576007402):0.04435347588):0.08
380065137,jacJac1:0.1438649666):0.04270536633):0.01663675397,(ochPri3:0.1256544445,oryCu
n2:0.07131655591):0.06535533418):0.009050428462,tupChi1:0.1191189141):0.003894252213):0.
01379370868,otoGar3:0.108738222):0.04299750653,(calJac3:0.02474184521,saiBol1:0.02096868
307):0.02784675729):0.0135408115,(chlSab1:0.007693724903,((macFas5:0.001292320552,rheMac
3:0.00713015786):0.002951690224,papHam1:0.005199240711):0.002049749893):0.01566263562):0
.007043113559,nomLeu3:0.01770384793):0.002187630666,ponAbe2:0.0164503644):0.00557232
7638,gorGor3:0.007765177171):0.001382639829,hg19:0.005957477577,panTro4:0.006721826689);

## Subset of master tree used for branch-site models for positive selection

(((((((((((lepWed1:0.02002160645,odoRosDi:0.02064385875):0.01734764946,musFur1:0.04613997497
):0.02373665057,felCat5:0.05020331605):0.03285617057,(((((((bosTau7:0.02168740723,oviAri3:0.0
3208995002):0.0662523666,(orcOrc1:0.006371664911,turTru2:0.01086552617):0.06014682602):0
.01216198069,susScr3:0.0796745271):0.006785823323,vicPac2:0.0747118422):0.02551888691,(c
erSim1:0.04977357056,equCab2:0.061454379):0.02510111297):0.00331214686,(myoDav1:0.139935
1075,pteAle1:0.08150735657):0.02385546003):0.002057771224):0.004845253848,(conCri1:0.123
9823369,sorAra2:0.2142153246):0.0235875333):0.01477733374,(((eleEdw1:0.1582970599,oryAfe
1:0.08326528894):0.008243787904,(loxAfr3:0.06812658238,triMan1:0.06198982615):0.02249945
29):0.03384011363,dasNov3:0.1342602666):0.01915406518):0.01425600689,(((((cavPor3:0.1116
757769,hetGla2:0.08588673524):0.07432515556,speTri2:0.08896424642):0.006291577528,((criG
ri1:0.08345878555,mm10:0.09708989861):0.08380065137,jacJac1:0.1438649666):0.04270536633)
:0.01663675397,oryCun2:0.1366718901):0.009050428462,tupChi1:0.1191189141):0.003894252213
):0.01379370868,otoGar3:0.108738222):0.04299750653,calJac3:0.0525886025):0.0135408115,(c
hlSab1:0.007693724903,rheMac3:0.01213159798):0.01566263562):0.01618571169,hg19:0.0059574
77577,panTro4:0.006721826689);

## Acknowledgements

We thank Dr. Andreas Pfenning and Dr. Dennis Kostka for helpful feedback, as well as Elysia Saputra, Weiguang Mao, and all members of the Clark and Chikina labs for helpful feedback. One or more of the authors of this article self-identifies as a member of the LGBTQ+ community. Funding for this research was provided through National Institutes of Health grants R01HG009299, R01EY030546, and U54 HG008540.

## Additional information

### Funding

| Funder | Grant reference number | Author |
| --- | --- | --- |
| National Institutes of Health | HG009299 | Amanda Kowalczyk |
| National Institutes of Health | EY030546 | Amanda Kowalczyk |

The funders had no role in study design, data collection and interpretation, or the decision to submit the work for publication.

### Author contributions

Amanda Kowalczyk, Conceptualization, Data curation, Software, Formal analysis, Validation, Investigation, Visualization, Methodology, Writing - original draft, Writing – review and editing; Maria Chikina, Conceptualization, Formal analysis, Supervision, Funding acquisition, Methodology, Writing

– review and editing; Nathan Clark, Conceptualization, Formal analysis, Supervision, Funding acquisition, Writing – review and editing

**Author ORCIDs**
Amanda Kowalczyk ![ORCID] http://orcid.org/0000-0002-9061-1336
Nathan Clark ![ORCID] http://orcid.org/0000-0003-0006-8374

**Decision letter and Author response**
Decision letter https://doi.org/10.7554/eLife.76911.sa1
Author response https://doi.org/10.7554/eLife.76911.sa2

## Additional files

### Supplementary files
• Transparent reporting form

• Source data 1. Supplementary data include all gene, noncoding, and enrichment results, noncoding region coordinates, and other source data shown in figures.

### Data availability
All data generated or analysed during this study are included in the manuscript and supporting file; Source Data files have been provided for all Figures. Code files are deposited in GitHub at https://github.com/nclark-lab/hairlessness copy archived at swh:1:rev:ccabcc3f723f42a16cd9e92fd02e53ec49a296b3.

The following dataset was generated:

| Author(s) | Year | Dataset title | Dataset URL | Database and Identifier |
|---|---|---|---|---|
| Clark N, Kowalczyk A, Chikina M | 2022 | Complementary evolution of coding and noncoding sequence underlies mammalian hairlessness | https://doi.org/10.5061/dryad.k98sf7m77 | Dryad Digital Repository, 10.5061/dryad.k98sf7m77 |

The following previously published dataset was used:

| Author(s) | Year | Dataset title | Dataset URL | Database and Identifier |
|---|---|---|---|---|
| Rhead B | 2010 | 100-way vertebrate genome alignment | http://hgdownload.cse.ucsc.edu/goldenpath/hg19/multiz100way/ | UCSC Genome Browser alignment, multiz100way |

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
