## [Editor Report]

Several mammal species, including dolphins, have evolved to be relatively "hairless". In this important work, Kowalczyk and colleagues scan the genomes of multiple species to identify genomic regions that appear to have evolved at a faster or slower evolutionary rate along hairless lineages. Using convincing analyses, they identify a number of protein-coding genes as well as noncoding regions that might explain how hairlessness evolved in mammals. This study is of interest to those investigating the development of the skin and its appendages as well as evolutionary biologists, especially those investigating instances of convergent evolution and those developing phylogenomic methods for genome comparisons.

---

## [Decision Letter]

**Decision letter after peer review:**

Thank you for submitting your article "Complementary evolution of coding and noncoding sequence underlies mammalian hairlessness" for consideration by *eLife*. Your article has been reviewed by 3 peer reviewers, and the evaluation has been overseen by a Reviewing Editor and Molly Przeworski as the Senior Editor. The following individual involved in the review of your submission has agreed to reveal their identity: Peter H Sudmant (Reviewer #1).

Essential revisions:

This is a strong manuscript that all three reviewers thought was interesting and well-done. The reviewers have provided a number of comments and suggestions. All the reviewers' comments and suggestions are constructive so we encourage you to carefully consider them in your revisions. In particular, please pay attention to:

1) Reviewer's #3 point about your assumption that species "use" the same genes to accomplish hairlessness. The reviewer has proposed a complementary analysis that would enable you to relax this assumption.

2) The numerous points raised by all three reviewers about clarity in your presentation of methods and results of the manuscript.

*Reviewer #2 (Recommendations for the authors):*

I do have some small comments and suggestions, some of which may be beyond the scope of the manuscript:

– It would seem that relaxed selection following hair reduction could eventually result in the total absence of a gene or CNE in the dataset. Can RERconverge handle gene or CNE deletions, and would these events be detected in the current approach?

– At the core of the paper is an argument that non-coding and coding genes are non-randomly contributing to the loss of hair. A hypothesis proposed here is based on pleiotropy, whereby the coding genes are under functional constraint and are limited to genes regulating hair physical structure and texture compared to regulatory regions near genes regulating developmental processes. Is there a way to assess this hypothesis with the data? For example, do the accelerated hair structure genes have more limited tissue-specific expression outside of hair or do they have fewer mammalian phenotypes by comparison to the set of genes associated with accelerated CNEs?

– Supplemental figure of correlation plots might be helpful for specific candidate genes mentioned in the text (FGF11, etc.), as it is unclear which combination of species is accelerated and driving the signature. This could be important when considering correlated integumentary appendage phenotypes (such as teeth). Further, are there any interesting mutations in these genes in any particular lineages? For coding genes, are there truncating or loss of function mutations associated with these genes?*Reviewer #3 (Recommendations for the authors):*

1) Is the word "permulation" necessary? The overall analysis is a permutation, as the goal is to permute the hairless phenotype across the species set. My understanding is that the Brownian simulation is only to account for phylogenetic relationships. Obviously, one cannot randomly permute phenotypes because that would assume all species are independent from each other. The Brownian simulation is a clever solution because it maps out how haired vs. hairless can be permuted while accounting for relatedness. But again… the analysis is fundamentally a permutation, just not sure a new word is necessary.

2) For coding regions, nonsynonymous substitutions are counted. I assume this is normalized by the number of nonsynonymous sites in a gene, but this is never mentioned. Or maybe it's not important to normalize since they are looking for relative rates of change across species. Please clarify.

3) It is not clear if/how branch lengths are taken into account. It seems that short branches would be noisier compared to long branches. Of course, this might be the opposite for a a rapidly evolving region, if long branches are more saturated with changes. Does branch length need to be taken into account? Or does it make a difference?

4) It is not clear how the authors chose the exact species included in Figure 1. They rely on the UCSC 100-way alignment. There are quite a few placental species that are in the UCSC 100 way alignment that were not included here.

5) Figure 1C and 1D: It is not clear how many genes were in each of those categories to begin with. Also, it's likely that many genes occur in more than one category, so it is just difficult to interpret the figures.

---

## [Author Response]

Essential revisions:This is a strong manuscript that all three reviewers thought was interesting and well-done. The reviewers have provided a number of comments and suggestions. All the reviewers' comments and suggestions are constructive so we encourage you to carefully consider them in your revisions. In particular, please pay attention to:1) Reviewer's #3 point about your assumption that species "use" the same genes to accomplish hairlessness. The reviewer has proposed a complementary analysis that would enable you to relax this assumption.

We have incorporated these analyses – please see the response to Reviewer #3’s point for details. See response to Reviewer #3’s point here:

Our analyses detect convergently evolving genomic elements associated with hairlessness for two reasons. First, species-specific analyses may detect genomic changes associated with any unique phenotypes in a particular species and it is difficult to distinguish which of those genomic changes are associated with hairlessness. Second, we are seeking genomic elements associated with hair growth in all mammals and species-specific adaptations will not be shared across all mammals.

Nevertheless, we conducted a complementary analysis to test for rate shifts specific to each hairless species compared to all of the non-hairless species. We then tested for enrichment of hair follicle genes among genes with significant rate shifts in different numbers of hairless species. For example, among all genes with significant rate shifts in at least one hairless species, is there an enrichment of hair follicle genes? Then, among all genes with significant rate shifts in at least two hairless species, is there an enrichment of hair follicle genes? Et cetera until we test for enrichment only in genes with rates shifts in all ten hairless species. As expected, the signal of enrichment gets stronger as more species share the rate shift (the “convergent signal”). This happens because the genes with shared rate shifts are more hair-specific than the genes with unshared rate shifts.

We also performed another analysis to test for enrichment of hair follicle genes among genes with significant rate shifts per hairless species. For example, in orca, are the genes with significant rate shifts enriched for hair follicle genes? To complement this analysis, we also repeated the procedure for non-hairless species for comparison. Only two of the ten hairless species show species-specific hair follicle enrichments, which indicates that most of the hairless species alone are insufficient to detect hair signal at all. Even among the two species with significant enrichment, there are thousands of total genes identified, many of which are likely related to other unique characteristics of those species other than hairlessness, and it is impossible to distinguish the hair-related genes from the other genes without additional information.

2) The numerous points raised by all three reviewers about clarity in your presentation of methods and results of the manuscript.

We have made numerous changes to the text to clarify presentation of our methods and results (as listed individually under relevant comments). We hope that these make our manuscript easier to understand.

Reviewer #2 (Recommendations for the authors):I do have some small comments and suggestions, some of which may be beyond the scope of the manuscript:– It would seem that relaxed selection following hair reduction could eventually result in the total absence of a gene or CNE in the dataset. Can RERconverge handle gene or CNE deletions, and would these events be detected in the current approach?

In the current approach, deleted elements (or elements missed due to things like alignment issues) are simply treated as missing values. The species set is subset to include only species that have sequence information on a per-element basis. For protein coding analyses, other tools could be used to identify missing genes and determine if missingness is associated with the hairless phenotype, but these analyses would require careful consideration of many factors such as gene copy number, genome quality, and defunctionalization criteria, and we therefore believe that redoing such analyses would be beyond the scope of this paper.

Considering CNE deletions would likely be a fruitful avenue to find gains and losses associated with hairlessness and other convergent phenotypes. However, this is challenging due to uncertainty about the source of missing sequence and lack of standard likelihood models for indels. We believe addressing these challenges is beyond the scope of this paper. Our analyses were performed on highly conserved potential regulatory regions, but a gain/loss analysis would require inclusion of less conserved regions, which would both be more difficult to identify and more sensitive to additional considerations like genome quality and defunctionalization criteria. Such analyses would further require cell type-specific analyses since activity of regulatory elements is highly varied across cell types.

– At the core of the paper is an argument that non-coding and coding genes are non-randomly contributing to the loss of hair. A hypothesis proposed here is based on pleiotropy, whereby the coding genes are under functional constraint and are limited to genes regulating hair physical structure and texture compared to regulatory regions near genes regulating developmental processes. Is there a way to assess this hypothesis with the data? For example, do the accelerated hair structure genes have more limited tissue-specific expression outside of hair or do they have fewer mammalian phenotypes by comparison to the set of genes associated with accelerated CNEs?

We agree that pleiotropy plays a huge role in evolutionary constraint, and we believe that constraint applies to both coding and noncoding regions. Just as genes have many functions, regulatory regions likely have different functions in different tissues and different developmental timepoints as well as in different species.

Some hair- and skin-related genes, like KRTs and KRTAPs, have relatively specific functions, but others have a wide range of functions, so as a rule, the genes we report are not particularly hair-specific. For example, FGF11 is a growth factor and GLRA4 appears to have a role in the central nervous system. However, since our knowledge of the full functionalities of all genes is incomplete, we do not think sufficient data exist to fully test the pleiotropy hypothesis because we simply do not know how pleiotropic the genes are, especially for lesser studied areas like hair follicle biology.

For noncoding regions, the dearth of data is even more notable. We would need to have tissue- and timepoint-specific regulatory activity across many hairy and hairless species to fully understand the regulatory machinery underlying the phenotype.

– Supplemental figure of correlation plots might be helpful for specific candidate genes mentioned in the text (FGF11, etc.), as it is unclear which combination of species is accelerated and driving the signature. This could be important when considering correlated integumentary appendage phenotypes (such as teeth). Further, are there any interesting mutations in these genes in any particular lineages? For coding genes, are there truncating or loss of function mutations associated with these genes?

We agree that plots for individual genes are important for interpretation and have included them as supplements to Figure 1. Note that all genes highlighted are supported by most, if not all, hairless species.

We also agree that analyzing defunctionalizing mutations in genes is very interesting and have added it for our top genes. The text now includes the following information regarding pseudogenes, and table 1 indicates pseudogenes in specific species.

“In fact, over half of our top genes from show evidence of pseudogenization, and therefore are defunctionalized, in one or more hairless species (Table 1) (W.K. Meyer et al., 2018).”

Reviewer #3 (Recommendations for the authors):1) Is the word "permulation" necessary? The overall analysis is a permutation, as the goal is to permute the hairless phenotype across the species set. My understanding is that the Brownian simulation is only to account for phylogenetic relationships. Obviously, one cannot randomly permute phenotypes because that would assume all species are independent from each other. The Brownian simulation is a clever solution because it maps out how haired vs. hairless can be permuted while accounting for relatedness. But again… the analysis is fundamentally a permutation, just not sure a new word is necessary.

We use the word permulation to match our cited publication that uses the word “permulation” to distinguish the specific combination of phylogenetic simulations and permutations from other strategies to generate null phenotypes. In practice, all three techniques (permulations, permutations, and phylogenetic simulations) are used in this context, so having a different word to distinguish permulations from unrestricted random shuffling (i.e. permutations) is something that we thought was important in communicating methods.

2) For coding regions, nonsynonymous substitutions are counted. I assume this is normalized by the number of nonsynonymous sites in a gene, but this is never mentioned. Or maybe it's not important to normalize since they are looking for relative rates of change across species. Please clarify.

The units on raw branch lengths are average number of substitutions per site, so a length correction already exists. (note that there are subsequent corrections of branch lengths to adjust for heteroskedasticity, normalize for average rate, etc., so the final branch lengths or RERs have different units from the raw branch lengths, but these are still accounting for sequence length).

3) It is not clear if/how branch lengths are taken into account. It seems that short branches would be noisier compared to long branches. Of course, this might be the opposite for a a rapidly evolving region, if long branches are more saturated with changes. Does branch length need to be taken into account? Or does it make a difference?

Indeed, longer branches generally show greater variability than shorter branches. This is a statistical anomaly known as heteroskedasticity and RERconverge corrects for it (demonstrated in Robust Method for Detecting Convergent Shifts in Evolutionary Rates, Partha et al. 2019). Branch lengths are converted to RERs (relative evolutionary rates) that are directly used to calculate associations, and the RERs include the statistical adjustments.

4) It is not clear how the authors chose the exact species included in Figure 1. They rely on the UCSC 100-way alignment. There are quite a few placental species that are in the UCSC 100 way alignment that were not included here.

The species chosen for Figure 1 were selected relatively arbitrarily to include all of the hairless species and a representative set of other species to demonstrate the breadth and structure of the tree. Note that this species selection was solely used for Figure 1A for ease of visualization – for analyses, the full set of 62 species was used. We have added a sentence to the Figure 1A legend to clarify that the species list was subset only for visualization purposes as follows:

“Note that all 62 species were included in analyses and only a subset are shown here for visualization purposes.”

5) Figure 1C and 1D: It is not clear how many genes were in each of those categories to begin with. Also, it's likely that many genes occur in more than one category, so it is just difficult to interpret the figures.

We have further investigated overlap in pathways highlighted in Figure 1 C and D. Note that Figure 1 Source Data 5 already included the information about pathway size, and we now additionally include Figure 1 Source Data 9 that reiterates that information as well as lists the number of genes shared between each pair of pathways. Note that we also provide a link to our pathway annotations so interested readers may further investigate or use the annotations themselves.

Supplements 2 and 3 to Figure 1 now show networks of pathway sizes and shared genes among pathways. Clusters were identified using a clustering algorithm through igraph that identifies neighborhoods based on the connectivity of the network. Families of pathways that clustered generally follow intuition about their relationships – for example, skin and hair pathways cluster together. Note that although pathways do overlap, they also have distinct genes. It would be surprising if related pathways were entirely distinct because genes often have many, sometimes quite divergent, functions. Overall, these findings are what we expect to see in these pathway annotations and do not affect our conclusions.